# Laser-Induced Intracellular Delivery: Exploiting Gold-Coated Spiky Polymeric Nanoparticles and Gold Nanorods under Near-Infrared Pulses for Single-Cell Nano-Photon-Poration

**DOI:** 10.3390/mi15020168

**Published:** 2024-01-23

**Authors:** Ashish Kumar, Bishal Kumar Nahak, Pallavi Gupta, Tuhin Subhra Santra, Fan-Gang Tseng

**Affiliations:** 1Department of Engineering and System Science, National Tsing Hua University, Hsinchu 300, Taiwan; raiash024@gmail.com (A.K.);; 2Department of Engineering Design, Indian Institute of Technology Madras, Chennai 600036, India; 3Department of Chemistry, National Tsing Hua University, Hsinchu 300, Taiwan; 4Research Center for Applied Sciences, Academia Sinica, Taipei 115, Taiwan; 5Institute of Nano Engineering and Microsystems, National Tsing Hua University, Hsinchu 300, Taiwan; 6Frontier Research Center on Fundamental and Applied Sciences of Matters, National Tsing Hua University, Hsinchu 300, Taiwan

**Keywords:** intracellular delivery, nanosecond-pulsed laser, Au-PNP, gold nanorods, photon-poration, LSPR, PVNB

## Abstract

This study explores the potential of laser-induced nano-photon-poration as a non-invasive technique for the intracellular delivery of micro/macromolecules at the single-cell level. This research proposes the utilization of gold-coated spiky polymeric nanoparticles (Au-PNPs) and gold nanorods (GNRs) to achieve efficient intracellular micro/macromolecule delivery at the single-cell level. By shifting the operating wavelength towards the near-infrared (NIR) range, the intracellular delivery efficiency and viability of Au-PNP-mediated photon-poration are compared to those using GNR-mediated intracellular delivery. Employing Au-PNPs as mediators in conjunction with nanosecond-pulsed lasers, a highly efficient intracellular delivery, while preserving high cell viability, is demonstrated. Laser pulses directed at Au-PNPs generate over a hundred hot spots per particle through plasmon resonance, facilitating the formation of photothermal vapor nanobubbles (PVNBs). These PVNBs create transient pores, enabling the gentle transfer of cargo from the extracellular to the intracellular milieu, without inducing deleterious effects in the cells. The optimization of wavelengths in the NIR region, coupled with low laser fluence (27 mJ/cm^2^) and nanoparticle concentrations (34 µg/mL), achieves outstanding delivery efficiencies (96%) and maintains high cell viability (up to 99%) across the various cell types, including cancer and neuronal cells. Importantly, sustained high cell viability (90–95%) is observed even 48 h post laser exposure. This innovative development holds considerable promise for diverse applications, encompassing drug delivery, gene therapy, and regenerative medicine. This study underscores the efficiency and versatility of the proposed technique, positioning it as a valuable tool for advancing intracellular delivery strategies in biomedical applications.

## 1. Introduction

Efficiently delivering foreign biomolecules into living cells while maintaining cell viability represents a significant challenge to their biological and therapeutic applications [1,2,3]. Various transfection techniques have been explored, encompassing viral vectors [4]; chemical methods, such as cationic lipids, polymers, and calcium phosphate complexes [5]; as well as physical methods, including electroporation [6,7], microinjection, and sonoporation [5]. Each method has its advantages and disadvantages, necessitating the ongoing innovation of intracellular delivery strategies. Chemical formulations, like polymers and cationic lipids, face issues related to targeting efficiency, endocytic degradation, and cell-specific requirements [8]. Meanwhile, viral vectors, although effective, pose challenges like mutagenicity, immune responses, toxicity, and high costs [9]. Physical methods, such as ultrasound and laser irradiation, offer alternatives, but they may not consistently achieve optimal efficiency [10]. Methods like microinjection stand out for their direct injection of micro/macromolecules into cells, but are limited by their delivery throughput and time-consuming process that necessitates the expertise of highly skilled operators [11,12]. Electroporation, another method, holds promise due to its ease of use and low toxicity [6,7]. Nevertheless, it is crucial to address concerns related to electrode-related issues that have the potential to compromise cell viability. In light of these challenges, there is a pressing need for innovative intracellular delivery strategies that can overcome these limitations and enhance the efficiency and safety of the process. In response to these challenges, researchers have explored photon-poration techniques, where coherent light creates transient pores in cell membranes for micro/macromolecule delivery [13]. Femtosecond laser pulses have achieved impressive cell-specific transfection efficiency, reaching up to 90% [14]. Continuous-wave lasers typically offer lower efficiency, less than 30% [14,15], while nanosecond laser pulses can achieve high efficiency, up to 90% [16,17].

In recent years, researchers have investigated the use of nanoparticles (NPs) to improve the efficiency of photonic energy coupling in cell poration [18]. Among these NPs, gold nanoparticles (GNPs) have garnered significant attention due to their unique properties, including their surface plasmon resonance, high chemical stability, biocompatibility, ease of surface bioconjugation, and low toxicity [18,19]. Various gold nanostructures, such as nanorods, nanoshells, nanocages, and half nanoshells, have demonstrated remarkable photothermal properties [19,20]. Due to the elevated surface plasmon resonance exhibited by GNPs, the application of high-intensity nanosecond laser pulses induces swift temperature escalation, causing the vaporization of the surrounding medium [21,22]. This process gives rise to the creation of plasmonic nanobubbles (PNBs), or photothermal vapor nanobubbles (PVNBs), along the surface of GNPs [23,24]. The subsequent collapse of these bubbles induces cavitation and stress waves, thereby promoting an efficient heat transfer from the GNPs to the surrounding environment [25]. The amount of heat produced is contingent upon various factors, encompassing particle size, particle aggregation, the intensity of the laser light, particle shape, the wavelengths used, and the pulse duration of the laser light. Nevertheless, in instances involving very short-lived PVNBs with a lifespan of less than 1 μs, the heat transferred from the GNPs to the adjacent medium is marginal, with the majority of the irradiated energy being transformed into mechanical energy [26,27,28]. Upon exceeding the threshold values of the cell membranes, the mechanical energy initiates the formation of transient pores, facilitating the gentle delivery of micro/macromolecules into the cells [13,26,29]. Researchers have made significant strides in this field. Xiong et al. used spherical GNPs to achieve a delivery efficiency of 85% for FITC-dextran with cell viability of approximately 90% at a very high laser fluence of 2 J cm^−2^ [30]. However, the high energy required for this approach has the potential to damage cells and is not energy efficient [29,30]. Therefore, there is still room for reducing the energy required for this process while simultaneously enhancing delivery efficiency and cell viability. Furthermore, most studies have employed visible wavelengths for cell poration [27,31]. It is widely acknowledged that the optimal penetration of electromagnetic radiation into biological tissue occurs within the near-infrared (NIR) range, typically spanning from 650 nm to 1350 nm [32,33]. The NIR range holds significant potential for in vivo applications, owing to its high biocompatibility and lower photothermal damage compared to the UV and visible ranges. The NIR range is particularly valuable for the advancement of photothermal therapy, making the achievement of photon-poration and biomolecular delivery within the NIR spectrum highly desirable.

In this study, we have developed a pioneering approach to intracellular micro/macromolecule delivery by leveraging the distinctive characteristics of Au-PNPs. This involved shifting the operating wavelength toward the NIR range at a lower fluence. Additionally, we conducted a comparative analysis of intracellular delivery using GNRs. Utilizing Au-PNPs as mediators, we employed nanosecond-pulsed lasers to achieve the highly efficient intracellular delivery of cargo molecules, while concurrently preserving high cell viability. When directing a laser pulse at the Au-PNPs, it instigated the generation of over a hundred hot spots per particle, resulting in heightened surface plasmon production across the Au-PNP surfaces compared to the spherical GNPs and GNRs. Consequently, the Au-PNPs exhibited superior energy transfer, culminating in the formation of PVNBs. This phenomenon facilitated a remarkably efficient intracellular delivery without inducing any deleterious effects on the cells. Our optimization endeavors underscored the efficacy of the approach at wavelengths of 670 nm and 1030 nm, achieving good delivery efficiencies and maintaining cell viability. This methodology not only significantly reduces energy requirements but also ensures heightened efficiency and cell viability. Consequently, this innovative development holds considerable promise for diverse applications, including drug delivery, gene therapy, and regenerative medicine.

## 2. Experimental Section

### 2.1. Materials and Reagents

The 530 nm carboxylic-modified polystyrene beads with no fluorescence, thiol-PEG-carboxylate [HSC_2_H_4_O (C_2_H_4_O) nCH_2_CO_2_H], ethanol (99.9%), chloroauric acid, CTAB, NaBH4, alkyltrimethylammonium chloride, NaOL, AgNO3, HAuCl4, HCl, and ascorbic acid were purchased from Sigma Aldrich, Maryland, USA. The PI dye (P1304MP, Invitrogen, Waltham, MA, USA) and Calcein-AM (C3099, Invitrogen, Waltham, MA, USA) were purchased from Life Technologies, Carlsbad, CA, USA. All reagents were of analytical grade and used without further purification. Milli-Q (mQ) water (Merck, Millipore Corp., Darmstadt, Germany) was used for all experimental purposes, and it was also used for Au-PNP and GNR syntheses and analyses. HCT-116 cell lines, HeLa cell lines, Neuro-2a (N2a) cell lines, and cell culture media (RPMI-1640 and DMEM, Dow Corning Co., Midland, MI, USA) were employed in this study.

### 2.2. High-Aspect-Ratio Gold-Coated Spiky Polymeric Nanoparticles (Au-PNPs) and Gold Nanorods (GNRs) Fabrication

Appendix A depicts the experimental setup employed for the creation of a densely packed monolayer of polystyrene beads (PSBs) at the water–air interface. The procedure commenced with the utilization of a circular glass dish filled with milli-Q (mQ) water. Precautions were taken to remove any scum or bubbles that might have accumulated at the water–air interface. Subsequently, a thin Teflon ring, measuring 16 cm in diameter, was carefully positioned at the water–air interface. This ring served as a means of confinement, delineating the area for the PSBs. The PSB suspension (500 μL) underwent concentration through centrifugation, undergoing three cycles of cleaning at 10,000 rpm for 20 min each. The purified PSBs were then mixed with a combination of ethanol and mQ water, with 50 μL of each component, and agitated for one minute. The resulting PSB solution was slowly introduced onto the surface of a nearly vertical conduit plate, as shown in Appendix A, using a syringe pump. This controlled injection (∼20 μL/min) minimized the risk of the harsh compression and irreversible aggregation of PSB particles due to van der Waals forces. The conduit plate itself was constructed using a silicon wafer, featuring a thin layer of thermal oxide. To enhance its affinity for mQ water, the plate underwent treatment with oxygen plasma. Once the Teflon ring was fully occupied by the PSBs, the setup was left undisturbed for 4–6 h to allow the PSB monolayer to settle on the water’s surface, as shown in Appendix A. Subsequently, the water was slowly drained. As the water evaporated, the densely packed monolayer of PSBs began to adhere to the surface of the silicon wafer, as depicted in Figure 1a and Appendix A. To induce additional modifications to the PSB monolayer and create a spiky surface, we subjected it to dry etching using the reactive ion etching (RIE) technique. The parameters used for this process included a power level of 200 W, an oxygen flow rate of 10 sccm, an etching time of 127 s, and a pressure of 60 mTorr. Subsequently, the deposition of titanium and gold (Ti/Au) with thicknesses of 10/20 nm was carried out via electron beam evaporation to create the final Au-PNP structure, as depicted in Figure 1a.

In contrast, the synthesis of GNRs involved a two-step process, as depicted in Figure 1b. Initially, for the preparation of the seed solution, 5 mL of a 0.5 mM chloroauric acid solution was mixed with 5 mL of a 0.2 M CTAB solution. To this mixture, 0.6 mL of freshly prepared NaBH4 at a concentration of 0.01 M was added, and diluted with mQ water to reach a final volume of 1 mL. The diluted NaBH4 solution was then carefully introduced into the Au-CTAB mixture while maintaining vigorous stirring at 1200 rpm. During this addition, the solution underwent a noticeable color change, transitioning from yellow to brownish-yellow. Stirring was halted after a 2 min interval. Following this step, the seed solution was left to age at room temperature for 30 min, rendering it ready for subsequent use. In the preparation of the growth solution, alkyltrimethylammonium chloride and NaOL were dissolved in 250 mL of warm water, at approximately 50 °C. The solution was then allowed to cool naturally until it reached a temperature of 30 °C. Next, a 4 mM AgNO3 solution was gently introduced into the flask, and the mixture was left undisturbed at 30 °C for 15 min. Following this, 250 mL of a 1 mM HAuCl4 solution was added to the mixture. Over a span of about 150 min, the solution transformed, becoming colorless, while being stirred at 700 rpm. To achieve the desired pH, a specific volume of HCl (37 wt. % in water, 12.1 M) was carefully added. After an additional 15 min of gentle stirring at 400 rpm, 1.25 mL of a 0.064 M ascorbic acid (AA) solution was introduced, followed by vigorous stirring for 30 s. Subsequently, a certain volume of the earlier-prepared seed solution was injected into the growth solution. The resulting mixture underwent further stirring for 30 s and was then left undisturbed at 30 °C for 12 h to facilitate the growth of nanorods. The final nanorod products were isolated through centrifugation at 6000 rpm for 25 min, followed by the removal of the supernatant. It is worth noting that no selective fractionation steps based on size or shape were undertaken during this process, ensuring the synthesis of gold nanorods in their diverse forms.

### 2.3. Au-PNPs and GNRs Modification with Thiol-PEG-Carboxylate

To enhance the colloidal stability of both the Au-PNPs and GNRs, a thiol-PEG-carboxylate (MW 3500 g/mol) was employed for encapsulation, resulting in the formation of Au-PNP-PEGs and GNR-PEGs. In this process, 1 mL of an aqueous suspension containing Au-PNPs and GNRs, both at a concentration of 34 μg/mL, was separately mixed with 1 mg of thiol-PEG-carboxylate. The pH of the solution was carefully adjusted to approximately 12.0 using a concentrated NaOH solution. The mixture underwent an overnight reaction, facilitating a ligand-exchange process, all while being gently stirred at 200 rpm. Subsequently, the reaction mixture was subjected to centrifugation three times for 15 min at a speed of 10,000 rpm, followed by dispersion in 1 mL of mQ water. The remarkable uniformity and stability of both the Au-PNPs and GNRs following this process provided strong evidence of the effective coating achieved with thiolate PEG, reinforcing their colloidal stability.

### 2.4. Characterization of Au-PNPs and GNRs

For the characterization of Au-PNPs and GNRs, a comprehensive set of advanced analytical methods was employed to reveal their properties and behavior. Firstly, the UV-Vis absorption characteristics of the Au-PNPs and GNRs were thoroughly analyzed using a JASCO V-670 spectrophotometer. A solution containing Au-PNPs and GNRs was separately placed within a cuvette, which was then positioned in the spectrophotometer chamber. Absorbance measurements were conducted across a wide range, up to 1400 nm. This method provided a thorough understanding of the nanoparticles’ light absorption capabilities across a diverse spectrum of wavelengths, offering valuable insights into their optical properties. Subsequently, a solution containing diluted Au-PNPs and GNRs was carefully drop-cast onto a silicon wafer measuring 0.5 × 0.5 cm^2^. The solution was allowed to gradually evaporate, leaving behind a concentration of nanoparticles on the wafer’s surface. To examine the prepared sample in detail, a high-resolution thermal field emission scanning electron microscope (JSM-7610F, JEOL Ltd., Tokyo, Japan) was utilized. This advanced microscopy technique allowed for the precise visualization of surface morphology, offering insights into the structural features of the nanoparticles.

### 2.5. Cell Culture

In our nanosecond-pulse laser-induced intracellular delivery experiments, three distinct cell lines were utilized: cervical cell lines (HeLas), colon cancer cell lines (HCT-116s), and mouse neuroblasts with neuronal and amoeboid stem cells (N2as). All these cell lines were originally sourced from the American Type Culture Collection (ATCC, Manassas, VA, USA). The HeLa cells were cultured in Dulbecco’s Modified Eagle’s Medium (DMEM), supplemented with 1% sodium pyruvate, 1% non-essential amino acids, and 1% penicillin-streptomycin (Gibco, Waltham, MA, USA). On the other hand, HCT-116 cells were cultured in McCoy’s 5a Medium Modified (Gibco, Waltham, MA, USA), supplemented with 10% fetal bovine serum (BI 04-001-1A) and 1% penicillin-streptomycin (Gibco, Waltham, MA, USA). For N2a cells, the initial culture involved the use of DMEM (Gibco, Waltham, MA, USA) along with 1% penicillin-streptomycin (Gibco, Waltham, MA, USA) to encourage the proliferation of neuronal cells without inducing differentiation. Prior to conducting the delivery experiments, the cells were subcultured and incubated overnight at a concentration of 3 × 10^5^ cells/mL to form a single layer. In the case of N2a cells, the addition of 10% FBS to the culture medium induced differentiation in the neuronal cells and facilitated axon growth. The cells were maintained under specific conditions, including 5% CO_2_ and a temperature of 37 °C. Sub-cultivation was performed at a ratio of 1:10 every 48 h. Before the commencement of the experiments, the cells were detached from the dish surface using trypsin (0.05% trypsin-EDTA, GIBCO), and subsequently suspended in the regular culture medium at a density of approximately 3 × 10^6^ cells per mL.

### 2.6. Fluorescence Microscopy

Fluorescence microscopy images were captured using an inverted fluorescence microscope (model IX73, Olympus, Tokyo, Japan) equipped with an epifluorescence attachment. In our photon-poration experiments, we employed 20× (0.4 NA, WD 3.2) and 40× (NA 0.6, WD 3.0–4.2) objectives for varying levels of magnification. Image acquisition was facilitated by using a color-cooled CCD camera (model MP5.0-RTVCLR-10-C) with a resolution of up to 2560 × 1920 pixels, and a flexible exposure time ranging from 1.6 milliseconds to 17.9 min. Fluorescence excitation was achieved using a 100-watt mercury lamp, accompanied by bright-field, dark-field, and differential interference contrast attachments. For narrowband UV excitation, we utilized an excitation filter (BP340-390), a dichroic beam splitter (DM410), and a barrier filter (BA538-IF). In cases of wideband blue excitation, an excitation filter (BP460-495), a dichroic beam splitter (DM505), and a barrier filter (BAS7S-IF) were employed. Lastly, for wideband green excitation, we utilized an excitation filter (BP560-550), a dichroic beam splitter (DM570), and a barrier filter (BA5S-IF) to suit our experimental requirements. The optical filters and splitters were procured from Tokyo, Japan.

### 2.7. Experimental Procedure

Before conducting the photon-poration experiment, cells were cultured in a Petri dish overnight. The specific micro and macromolecules of interest were introduced into the cell culture media prior to laser exposure. Following a 10 min incubation period, PI dye and Dextran 3000 MW were separately added to the cell culture. We employed an NT342B tunable nanosecond-pulse laser (EKSPLA, Vilnius, Lithuania) with a laser spot size of 6 mm to irradiate the samples. The laser output featured a pulse width of 5 nanoseconds at a frequency of 10 Hz. The laser fluence was adjusted within a range of up to 50 mJ/cm^2^. To ensure thorough coverage, the Petri dish area was continuously scanned by moving the sample using the stage.

### 2.8. Image Analysis

Fluorescence images capturing the delivery of both micro and macromolecules, as well as cell viability, were processed using ImageJ software to determine cell counts. To calculate delivery efficiency, we counted the number of cells that exhibited fluorescence due to PI dye or Dextran 3000 MW delivery. This count was divided by the total number of cells and multiplied by 100%. For cell viability assessment, a similar procedure was followed after staining with Calcein-AM and subsequent imaging. This process was replicated for each experimental condition using sets of fluorescence and bright-field images. Data were averaged from three images obtained under the same experimental conditions.

### 2.9. Statistical Analysis

Graphs were analyzed using Origin 2023b software for curve-fitting procedures. The data are presented as means ± standard error of the mean, derived from a minimum of three independent biological experiments.

## 3. Results and Discussion

### 3.1. Au-PNP-PEG- and GNR-PEG-Mediated Intracellular Delivery

In this study, we have meticulously devised an experimental procedure with which to achieve intracellular delivery mediated by high-aspect-ratio gold-coated spiky polymeric nanoparticles (Au-PNPs) and gold nanorods (GNRs) through the application of nanosecond-pulsed laser irradiation. Figure 1 provides an encompassing overview of the experimental procedure, focused on the intracellular delivery facilitated by the Au-PNPs using a nanosecond-pulsed laser. To initiate the process, the cells were cultured overnight within a controlled incubator environment at 37 °C and 5% CO_2_. Subsequently, thiol-PEG-carboxylate modified nanoparticles (Au-PNP-PEGs) were introduced into the cell culture dish, allowing for an incubation period of one hour prior to the experiment (Figure 1a). Unbound Au-PNPs were subsequently washed from the cell surface using phosphate-buffered saline (PBS). Following this, cell-impermeable cargo molecules (micro/macro) were introduced into the culture dish immediately before the experiment (Figure 1b).

As the nanosecond-pulsed laser irradiated, the Au-PNPs affixed to the cell membrane (Figure 1c) initiated an electromagnetic field enhancement, particularly near the nanosized corrugated edges and the smooth surface of the nanoparticles [21,22]. This enhancement, analogous to a lightning-rod effect, led to the formation of optical hotspots. Resonantly excited optical hotspots produced localized heat, resulting in a rapid temperature increase and the formation of PVNBs encompassing the Au-PNP–cell membrane interface [23]. These PVNBs experienced rapid growth, coalescence, and eventual collapse, creating an explosion that induced a robust fluid flow at the Au-PNP–cell membrane interface [34,35]. After the PVNBs collapse, the resulting cavitation and stress wave facilitated the transfer of heat from the Au-PNPs to the localized region of the cell membrane [36]. The nanosecond-pulsed laser exposure, characterized by a pulse duration of approximately 5 ns and a repetition rate of 10 Hz, generated PVNBs with exceedingly short lifetimes (~1 μs). This distinctive characteristic minimized the heat transfer from the Au-PNPs to the cell membrane. Consequently, the majority of the irradiated energy from the Au-PNPs was transformed into mechanical energy [26,27,28], leading to the disruption of the plasma membrane and the creation of transient membrane pores, enabling the delivery of micro/macromolecules from the external environment into the cell (Figure 1d) [13,26,29]. Once the micro/macromolecules were successfully delivered, the cell membrane resealed, thereby preserving cell viability (Figure 1e). This identical procedure was employed for the delivery of micro/macromolecules using GNRs. An illustration depicting the GNR-mediated delivery of impermeable micro/macromolecules can be found in Appendix A. The formation of transient pores on the cell membrane depended on factors such as laser fluence and the concentrations of the Au-PNPs/GNRs. For a detailed understanding of the fabrication process for the Au-PNPs and GNRs, please refer to Figure 1 and Appendix A. Following the fabrication process, we conducted surface modifications on the particles, establishing a covalent bond between the Au-PNPs/GNRs and polyethylene glycol (PEG) through the introduction of thiol-PEG-carboxylate. This modification was essential to enhance the adhesion of these particles to the cell membrane. The SEM imaging, as demonstrated by Tuhin et al., confirmed that the GNPs effectively adhered to the cell membrane surface, with an average attachment of 50–60 GNPs to each cell.

A thorough assessment of the optical and morphological characteristics of both the Au-PNPs and GNRs was meticulously conducted, employing UV-Vis absorption spectroscopy and scanning electron microscopy (SEM). As demonstrated in Figure 2a,b, the spectroscopic analysis unveiled distinct peaks at 538 nm and 670 nm for the Au-PNPs, while the GNRs exhibited prominent peaks at 560 nm and 1030 nm. These peaks are attributed to interband electronic transitions, as well as the surface plasmon resonance at the gold surface [37,38]. The SEM images offered a revealing glimpse into the morphological characteristics of the fabricated Au-PNPs with an average diameter of 270 nm and boasting over a hundred hotspots (Figure 2c). In contrast, the GNRs possessed an aspect ratio (A) of 4, featuring a length of approximately 200 nm and a width of around 50 nm (Figure 2d).

Appendix A presents compelling evidence that the distinct exposure times during the RIE led to observable modifications in the morphology of the Au-PNPs. This observation underscores the significant influence of exposure duration on the structural characteristics of the nanoparticles, emphasizing the sensitivity of the etching process and its capacity to shape the final morphology of the Au-PNPs. This nuanced understanding of the relationship between exposure times and morphological changes is pivotal for tailoring and optimizing the fabrication processes for Au-PNPs for various applications. Also, the plasmon peak position of these Au-PNPs can be tailored by adjusting the aspect ratio through various techniques, such as dry etching, thin-film coating, or wet-chemistry synthesis. Au-PNPs, characterized by their highly corrugated surface and spiky architecture, enhance nanolocalized plasmonic effects, thereby contributing to efficient photothermal conversion and targeted intracellular delivery. With a superior absorption cross-section, Au-PNPs stand out as a promising candidate for effective light absorption. However, the synthesis process for its corrugated surface presents challenges in terms of reproducibility and scalability. In contrast, GNRs offer versatility in their optical properties, allowing for tunable absorption wavelengths [39,40]. Surface modifications enable targeted cellular uptake, enhancing the specificity of intracellular delivery. Nevertheless, challenges persist, including aspect ratio-dependent properties requiring careful optimization, and the need for a thorough investigation into potential cytotoxic effects and clearance to ensure biocompatibility [41,42]. The primary focus of this study centers on assessing the impact of various experimental parameters, particularly at excitation wavelengths in the near-infrared regions (670 nm and 1030 nm).

To provide empirical support for the pivotal role of PVNBs in cellular transfection at these specific wavelengths, our previous investigation presented the outcomes of electromagnetic simulations [13,29,43]. These simulations were conducted through a rigorous 3D finite element analysis using Comsol Multiphysics software 5.2. In these simulations, the model particles were configured with core–shell structure, where the core consisted of polystyrene and the shell was composed of gold material. The results of the simulations clearly illustrated that the plasmonic electromagnetic field enhancement at the edges of the Au-PNPs was approximately 4–5 times higher compared to that of the gold nanospheres (GNSs) [13,29,44]. Additionally, previous simulations of GNRs have shown electric field enhancements at 1030 nm for various aspect ratios (1 <= Aspect ratio (A) <= 5) [43]. These simulations have provided clear evidence that GNRs generate considerably higher temperatures compared to GNSs, and Au-PNPs generate even higher temperatures than GNRs when exposed to identical light intensities. For instance, if we consider a sphere with a diameter of 180 nm, its volume would be 3.05 × 10^−3^ μm^2^, significantly larger than the volume of a nanorod with dimensions of D = 50 nm and A = 5, which is 0.46 × 10^−3^ μm^2^. However, the absorption cross-section for the nanorod was notably higher than that for the sphere, resulting in a much greater temperature increase for nanorods compared to nanospheres under an equivalent light intensity [43]. In contrast, the Au-PNPs exhibited superior enhancement due to its spiky surface, which provided a higher absorption cross-section compared to the GNSs and GNRs [29,43,44]. This led to nanolocalized plasmonic enhancement in the proximity of the corrugated surface tips and inter-spike nanospaces, creating plasmonic hotspots and enhancing Joule heating. Consequently, the temperature of the gold nanoparticle increased above its initial temperature and diffused into the surrounding medium. A noticeable redshift of the localized surface plasmon resonance (LSPR) wavelength from the visible to the infrared, along with an augmented absorption cross-section, were observed, as the aspect ratio (L/D), i.e., the length (L) of the GNR, increased while maintaining a constant diameter (D) of 50 nm [43]. This is in line with Lombard et al.’s estimation that cavitation bubbles initiate generation when the lattice temperature of gold at the gold–water interface surpasses the spinodal temperature (Ts ∼550 K) [45]. Hence, we anticipated the formation of PVNBs around the nanoparticle–water interface, with the spiky-surface particles exhibiting more prominent bubble generation due to their considerably higher temperature rise compared to the smooth-surface particles. In the course of these procedures, the occurrence of bubble generation at the gold–water interface coincided with instances where the temperature of the gold lattice exceeded 550 K [45,46]. Loganathan et al. provided an experimental visualization of bubble formation and its temporal dynamics at each step following pulse excitation [46]. The cumulative findings substantiate the efficacy of the spiky surface of the Au-PNPs for engendering nanolocalized plasmonic enhancement, leading to heightened Joule heating and, consequently, more conspicuous bubble generation. Furthermore, these results affirm the pivotal role of PVNBs in the cellular transfection process, providing insights into the nuanced dynamics of plasmonic nanoparticles in facilitating intracellular delivery.

### 3.2. Impact of Au-PNPs and GNRs Concentration, Exposure Time, and Laser Fluence on Delivery Efficiency and Cell Viability of Cells

In our systematic investigation, we explored the impact of the Au-PNPs and GNRs concentration, exposure time, laser fluence, and wavelength on the efficiency of delivering PI dye into cells, as well as cell viability. We aimed to identify the optimal conditions for this process. Our experiments revealed that the highest efficiency and cell viability for PI dye uptake was achieved with a concentration of 34 μg/mL of both Au-PNPs and GNRs. Laser exposure at 27 mJ/cm^2^ fluence for 30 s, using a 10 Hz pulse frequency at both 670 nm and 1030 nm, was found to be optimal for HeLa, HCT-116, and N2a cells. Beyond this concentration, increasing the particle concentration above 34 μg/mL for both the Au-PNPs and GNRs resulted in decreased delivery efficiency and reduced cell viability, as shown in Appendix A. At higher concentrations, specifically 54 μg/mL, we observed increased debris production and a substantial reduction in cell viability and delivery efficiency, which was evident from the bright-field images and Calcein-AM staining (Appendix A). Importantly, our findings demonstrate that when pulsed lasers were applied to cells without the presence of Au-PNPs, no delivery of PI dye into the cells was observed (Appendix A). This highlights the critical role of appropriate particle concentration for intracellular delivery. Furthermore, to assess toxicity, we conducted tests with HeLa cells, exposing them to a pulse laser on the first day with the Au-PNPs. Subsequently, we evaluated cell viability over different days and found that the majority of the cells remained viable after three days, as depicted in Appendix A. We examined the delivery efficiency and cell viability for the HeLa cells as a function of laser-exposure time, employing a laser fluence of 27 mJ/cm^2^ at 670 nm and 1030 nm, as shown in Appendix A. The results indicate that increasing the exposure time up to 30 s leads to peak delivery efficiencies of approximately 96% and 91%, as well as cell viabilities of about 99% and 92% for the Au-PNPs and GNRs, respectively.

However, beyond this exposure time, both delivery efficiency and cell viability exhibited a decline. Furthermore, we investigated the impact of laser fluence on intracellular delivery efficiency and cell viability for the HeLa cells. This analysis considered two wavelengths: 538 nm and 670 nm for the Au-PNPs and 560 nm and 1030 nm for the GNRs, each with a 27 s laser exposure (Figure 3). Parallel experiments were also carried out with HCT-116 and N2a cell lines at 670 nm and 1030 nm, as shown in Appendix A. Our results revealed the highest delivery efficiency, which reached 88%, 79%, and 92%, along with 98% cell viability, for the respective wavelengths of 538 nm and 560 nm. Beyond this fluence, both the delivery efficiency and cell viability exhibited a decline, as depicted in Figure 3. At 670 nm and 1030 nm, the maximum delivery efficiency was 96% and 91%, with cell viability reaching 99% and 92% at the same laser fluence (27 mJ/cm^2^), as shown in Figure 3. These findings underscore the finding that delivery efficiency and cell viability were notably improved at 670 nm and 1030 nm compared to 538 nm and 560 nm. To validate these results, we also tested a non-resonant wavelength at 600 nm, where the PI dye delivery was not observed, indicating a lack of cell penetration, as seen in Appendix A. Some cells exhibited red fluorescence, suggesting dead cells without intact cell cytosols (Appendix A). The optimum laser irradiation conditions were determined to be 27 mJ/cm^2^ for all the corresponding wavelengths. However, at higher laser fluences, specifically 50 mJ/cm^2^, a significant proportion of the cells became non-viable, with only about 4–7% of the cells remaining viable, highlighting the critical importance of fluence control in these experiments, as evidenced in Appendix A. Furthermore, it is essential to acknowledge that prior studies on cell photon-poration induced by PVNBs often required significantly higher fluences to achieve successful cell poration.

### 3.3. Intracellular Delivery of Molecular Dye (PI) and Dextran 3000

In our experiments, we employed propidium iodide (PI) dye as a molecular probe with which to evaluate intracellular delivery efficiency. PI dye, which is inherently impermeable to cells, can enter the cytosol only after the physical disruption of the cell membrane [7,29,47]. Prior to activating the nanosecond-pulsed laser, we introduced the PI dye at a concentration of 50 μL/mL, as demonstrated in Appendix A**,** which illustrates the delivery efficiency and viability at varying dye concentrations. To assess cell viability, we introduced cell-permeable Calcein-AM dye after two hours of laser exposure. Calcein-AM undergoes hydrolysis within live cells, producing a green fluorescence [6,48].

Figure 4 showcases the successful delivery of the PI dye into the HeLa, HCT-116, and N2a cells, while maintaining their high viability. Figure 4c,g,k depict the successful PI dye delivery into these cells, Figure 4b,f,j illustrate the corresponding live cells after the PI dye delivery, and Figure 4d,h,l present the merged images showing the PI dye delivery alongside live cell imaging at a 670 nm wavelength. The effectiveness of the Au-PNPs and GNRs for delivering the PI dye into cells was further confirmed by performing area-restricted pulse laser exposure on cells with Au-PNPs and GNRs, as shown in Appendix A. These results reveal that the PI dye was exclusively delivered within the exposed area in the HeLa cells, and all the cells remained viable (Calcein-AM staining) after delivery, regardless of the exposure or non-exposure areas. Similar results were observed for the HCT-116 and N2a cells. The efficient intracellular delivery of larger cargo is crucial in biological research and clinical therapy [2,49]. To evaluate the performance of the Au-PNPs and GNRs for larger biomolecular delivery, we conducted photon-poration experiments using dextran 3000 MW with the HeLa, HCT-116, and N2a cells. Dextran 3000 MW is a cell-impermeable polysaccharide with a higher molecular weight than the PI dye, making it challenging to deliver inside a cell. The experimental procedure closely mirrors the previous ones, and Figure 5c,g,k illustrate the fluorescence images of the dextran 3000 MW delivery and the assessment of cell viability in the HeLa, HCT-116, and N2a cells at 27 mJ/cm^2^ laser fluence, with a 670 nm laser excitation at a 10 Hz pulse frequency for 30 s. Calcein-AM staining of the live cells, shown in Figure 5b,f,j reveals that the majority of cells remained viable after the dextran 3000 MW delivery, and the merged images shown in Figure 5d,h,l illustrate the successful delivery of the dextran 3000 MW into the cells. In the context of small-molecule (PI dye) delivery, an augmented delivery speed or quantity signifies a collective influence on both pore sizes and numbers [50]. This observation implies an increased total number of pores and pore areas. Conversely, for larger molecules, such as Dextran, a greater necessity for larger nanoholes in the cellular delivery is indicated. Consequently, a higher value in this context corresponds to an augmentation resulting in larger pore sizes.

Should Au-PNPs exhibit elevated delivery efficacy for both small and large molecules, it would imply a heightened capacity to generate larger pores, resulting in an accumulation of larger total pore sizes. The data presented in Appendix A provide clear evidence of higher delivery rates and viability under identical laser irradiation conditions. These results collectively indicate that Au-PNPs facilitate superior PVNB generation and larger pore formation in comparison to GNRs. This conclusion is drawn from the observed enhanced delivery efficiency and viability outcomes, substantiating the notion that Au-PNPs excel at both small- and large-molecule delivery, thereby contributing to increased PVNBs and larger pore dimensions compared to GNRs. In Figure 6a, we present the delivery efficiency for both PI dye and Dextran 3000 MW, as well as cell viability, at a wavelength of 670 nm using Au-PNPs. We achieved a maximum delivery efficiency of 96% for the HeLa cells, coupled with an impressive cell viability of 99%. Consistently, in the cases of the HCT-116 and N2a cells, the delivery efficiency of the PI dye was a robust 93%. Additionally, for the Dextran 3000 MW, we achieved a noteworthy delivery efficiency, measuring at 91% for the HCT-116 and 95% for the N2a cells. Moreover, the cell viability remained remarkably high, with the HCT-116 and N2a cells registering at 96% and 97%, as well as at 94% and 96%, respectively. Figure 6b portrays the outcomes of the GNR-mediated photon-poration at a wavelength of 1030 nm. Here, we attained a maximum delivery efficiency of 91% for the HeLa cells, complemented by a cell viability of 92%. Similarly, for the HCT-116 and N2a cells, we consistently observed the favorable delivery efficiency of the PI dye, measuring 92% and 91%, respectively. This study’s cumulative findings affirm the efficacy of Au-PNPs’ spiky surface at inducing a nanolocalized plasmonic enhancement, leading to the generation of photothermal vapor nanobubbles (PVNBs). This enhancement plays a pivotal role in cellular transfection, demonstrating a high delivery efficiency for both smaller (PI dye) and larger (Dextran) molecules when compared to the smooth-surfaced GNRs, as depicted in Figure 6. The observed disparity in the delivery efficiency between the PI dye and Dextran can be attributed to their size difference. Smaller molecules (PI dye) exhibit higher diffusion rates, facilitating a rapid penetration of cellular barriers [51]. Conversely, the larger size of the Dextran molecules introduces steric hindrance, impeding their efficient navigation through the cellular environment [52]. The findings contribute valuable insights into the interplay between nanoparticle morphology, molecular size, and intracellular transportation dynamics.

The corresponding cell viability stands at 91% and 90%. When dealing with the larger cargo of Dextran 3000 MW, we noted a slightly lower, yet still remarkable, delivery efficiency, which was 91% for the HCT-116 and 88% for the N2a cells. In terms of cell viability, it remained notable, reaching 94% and 92% for the HCT-116 and N2a cells, respectively. Our observations emphasize that cell viability was notably higher when utilizing laser wavelengths of 670 nm and 1030 nm, compared to 538 nm and 560 nm, as depicted in Appendix A. At 538 nm, the best results yielded 88% delivery efficiency and 92% cell viability for the HeLa cells. Meanwhile, the HCT-116 and N2a cells exhibited delivery efficiencies of 85% and 91%, accompanied by cell viabilities of 87% and 92%, respectively. Upon a meticulous examination and validation of the evidence presented in this manuscript, it becomes evident that cellular viability is significantly higher when utilizing Au-PNPs compared to GNRs. Furthermore, enhanced cellular viability was notably observed with laser wavelengths in the near-infrared (NIR) spectrum, specifically at 670 nm and 1030 nm. This underscores the advantageous impact of employing Au-PNPs and optimizing the photoporation wavelength within the NIR range for improved cell viability, and underscores the robustness of Au-PNP-assisted delivery for both smaller and larger cargos, highlighting its potential for versatile applications in intracellular studies.

### 3.4. Cell Viability Test on Different Days

This study involved the evaluation of cell proliferation and viability across three distinct cell lines: HeLa, HCT-116, and N2a, at three different time points, specifically 2 h, 24 h, and 48 h. To assess cell viability, we employed a CCK-8 colorimetric assay, both before and after laser exposure, using Au-PNPs and GNRs with the aforementioned cell lines. The enhancement of viable cells post-laser irradiation was evident from the increased absorbance readings at an optical density (OD) of 450 nm during the 2 h, 24 h, and 48 h incubation periods. These absorbance values were subsequently transformed into percentages of cell viability, as depicted in Figure 7. Following laser irradiation with Au-PNPs, a substantial increase in cell viability was observed after a 24 h incubation period. The specific measurements revealed cell viabilities of 99%, 96%, and 97% for the HeLa, HCT-116, and N2a cells, respectively, with accompanying deviations of 5.4%, 6.1%, and 4.6% (Figure 7a). In contrast, the GNR-mediated laser irradiation resulted in cell viabilities of 92%, 91%, and 90%, with deviations of 6.8%, 4.8%, and 6.3% for the HeLa, HCT-116, and N2a cells, respectively (Figure 7b). The count of viable cells displayed a continuous increase at the 24 h and 48 h marks during the incubation period, with no evident signs of cytotoxicity, even after an extended incubation of 48 h (Figure 7a,b). It is crucial to highlight that there was only a minimal occurrence of cell death, ~2–3%, following laser exposure. Contrary to this, the cell population and proliferation efficiency demonstrated a consistent upward trajectory as the incubation period extended, underscoring the enhanced growth potential associated with longer incubation durations. Furthermore, the cells displayed a well-spread morphology and a notably increased cell count, providing clear evidence of augmented growth during the extended incubation period. These findings underscore the overall robustness and safety of the laser-assisted methods employed in this study, showcasing their capacity to promote cell viability and growth. Additionally, these outcomes emphasize the potential translational applications of these techniques to the realms of drug delivery and gene therapy. For drug delivery, the demonstrated ability to achieve a targeted and efficient intracellular delivery suggests breakthroughs in therapeutic outcomes with minimized side effects. Similarly, for gene therapy, the precise delivery of genetic material could be advanced, promising increased precision and efficacy in manipulating cellular processes. The translational potential of this study is scientifically grounded in the unique features of the employed technique, providing a robust rationale for its application to these biomedical domains. Consequently, this investigation serves as a foundational step towards bridging the gap between fundamental research and impactful translational applications, offering the possibility of transformative advancements in drug delivery and gene therapy.

## 4. Conclusions

Our study introduces a novel approach utilizing Au-PNP-PEGs and GNR-PEGs for photon-poration through nanosecond laser pulses, aiming to enhance intracellular delivery efficiency while maintaining robust cell viability. The distinctive asymmetry of Au-PNPs results in over a 100-fold increase in the number of hotspots compared to GNPs and GNRs, harnessing more laser energy at 670 nm through plasmon resonance. This process generates localized heat, forming PVNBs that induce a vigorous fluid flow at the interface of the NPs and the cell membrane. The ensuing transient membrane pores facilitate the gentle transfer of exogenous cargo from the extracellular to the intracellular milieu.

Our study substantiates the effectiveness of the proposed method for achieving the successful intracellular delivery of PI dye and dextran 3000 MW with high efficiency and cell viability across diverse cell types, including cancer cells and neuronal cells. The application of this technique to intracellular delivery in these diverse cells, utilizing a reduced laser fluence (27 mJ/cm^2^) at wavelengths of 670 nm and 1030 nm, in conjunction with low nanoparticle concentrations (34 μg/mL), yielded a substantial improvement in delivery efficiency, attaining levels of up to 96%, accompanied by an impressive 99% cell viability. Moreover, sustained high cell viability, ranging from 90% to 95%, was observed even 48 h post laser exposure. These results underscore the robustness and versatility of the method across various cell lines, and hold significant promise for advancing intracellular delivery strategies in biomedical applications. Future intracellular delivery techniques may leverage nanotechnological innovations, focusing on refining smart nanocarriers for precise targeting and controlled release. Advanced imaging techniques, incorporating real-time and non-invasive modalities, offer enhanced insights, while integrating artificial intelligence and machine learning can optimize delivery strategies. Synthetic biology applications, such as engineering cells for heightened receptivity or creating synthetic organelles, are also promising. Additionally, intracellular sensing systems, biohybrid carriers, and advancements in microfluidics and lab-on-a-chip technologies could contribute to more sophisticated and efficient intracellular delivery methodologies. Designing nanoparticles capable of delivering larger cargo molecules with high cell viability and efficiency at low laser fluences is crucial for future advancements in this field.

## Data Availability

Data are contained within the article and Appendix A.

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
