# Peer review of "Laser-Induced Intracellular Delivery: Exploiting Gold-Coated Spiky Polymeric Nanoparticles and Gold Nanorods under Near-Infrared Pulses for Single-Cell Nano-Photon-Poration"

_micromachines, 2024, doi:10.3390/mi15020168_

Round 1

Reviewer 1 Report

Comments and Suggestions for Authors

In this study, nanosecond pulsed laser irradiation was appliet to realize intracellular delivery of Au-PNP and GNR.

Pulse width, light energy, and nanoparticle/nanorod concentration are optimized to maximize delivery efficiency and cell viability.

Authors mentioned “Hence, we an-ticipated the formation of PVNB around the nanoparticle-water interface, with spiky sur-face particles exhibiting more prominent bubble generation due to their considerably higher temperature rise compared to smooth surface particles.” There is no significant difference between Au-PNPs and GNRs in terms of cell viability. I am interested in whether there is a significant difference between Au-PNP and GNR in terms of the amount of PVNB generated.

Section 3.3; The alphabets in the diagram below are incorrect, so please correct them.

Figures 4(a), 4(d), and 4(g) depict the successful PI dye delivery into these cells, Fig-ures 4(b), 4(e), and 4(h) illustrate the corresponding live cells after PI dye delivery, and Figures 4(c), 4(f), and 4(i) present merged images showing PI dye delivery alongside live cell imaging at 670 nm wavelength.

Figures 5(a), 5(d), and 5(g) illustrate fluorescence images of dextran 3000 MW delivery and the assess-ment of cell viability in HeLa, HCT-116, and N2a cells at 27 mJ/cm2 laser fluence with 670 nm laser excitation at a 10 Hz pulse frequency for 30 seconds.

Calcein-AM staining of live cells in Figures 5(b), 5(e), and 5(h) revealed that the ma-jority of cells remained viable after dextran 3000 MW delivery, and merged images in Figures 5(c), 5(f), and 5(i) illustrated the successful delivery of dextran 3000 MW into cells.

Author Response

Dear Reviewer,

I trust this message finds you well. I am writing to express my sincere appreciation for the time and effort you dedicated to reviewing our manuscript titled "Laser-Induced Intracellular Delivery: Exploiting Gold-Coated Spiky Polymeric Nanoparticles and Gold Nanorods under Near-Infrared Pulses for Single Cell Nano-Photonporation". Your insightful comments and constructive feedback have been invaluable in shaping the quality and rigor of our work.

Your expertise and thorough evaluation have significantly contributed to the enhancement of our manuscript. We have carefully considered each of your suggestions and have made the necessary revisions to address the concerns raised. We believe these changes have strengthened the overall merit of the paper.

We are truly grateful for your commitment to maintaining the standards of academic excellence. Your dedication to the peer review process is instrumental in advancing scholarly discourse.

Should you have any additional comments or require further clarification, please do not hesitate to reach out. We welcome the opportunity to continue refining our work based on your expertise.

Once again, thank you for your time, diligence, and invaluable contribution to our research. We look forward to the possibility of receiving your final feedback.

Reviewer 2 Report

Comments and Suggestions for Authors

This manuscript presents a novel technique for intracellular delivery of micro/macromolecules at the single-cell level using laser-induced nano-photonporation. The study demonstrates the potential of gold-coated spiky polymeric nanoparticles (Au-PNP) and gold nanorods (GNR) as mediators for efficient intracellular delivery under near-infrared pulses. The article highlights the importance of shifting the operating wavelength towards the NIR range to enhance intracellular delivery efficiency and cell viability. Overall, the article provides valuable insights into the development of non-invasive intracellular delivery techniques for various applications in biology and medicine.

Comments:

1         The article is well-written and the research is clearly presented. However, more detailed information about the materials used, such as their specific properties and synthesis methods, would enhance the article.

2         The results section could benefit from more in-depth analysis and discussion on the observed intracellular delivery efficiency and cell viability.

3         It would be interesting to compare the Au-PNP and GNR mediators in more detail, especially in terms of their respective advantages and disadvantages for intracellular delivery.

4         The discussion section should also explore the potential applications of this technique beyond basic research, such as in drug delivery or gene therapy.

5         The article could benefit from including more graphical representations, such as graphs or schematics, to visually illustrate key points and concepts.

6         Finally, it would be useful to consider adding a section on potential future research directions or technological advancements that could further improve intracellular delivery techniques.

Author Response

Dear Reviewer,

I trust this message finds you well. I am writing to express my sincere appreciation for the time and effort you dedicated to reviewing our manuscript titled “Laser-Induced Intracellular Delivery: Exploiting Gold-Coated Spiky Polymeric Nanoparticles and Gold Nanorods under Near-Infrared Pulses for Single Cell Nano-Photonporation”. Your insightful comments and constructive feedback have been invaluable in shaping the quality and rigor of our work.

Your expertise and thorough evaluation have significantly contributed to the enhancement of our manuscript. We have carefully considered each of your suggestions and have made the necessary revisions to address the concerns raised. We believe these changes have strengthened the overall merit of the paper.

We are truly grateful for your commitment to maintaining the standards of academic excellence. Your dedication to the peer review process is instrumental in advancing scholarly discourse.

Should you have any additional comments or require further clarification, please do not hesitate to reach out. We welcome the opportunity to continue refining our work based on your expertise.

Once again, thank you for your time, diligence, and invaluable contribution to our research. We look forward to the possibility of receiving your final feedback.
